# Targeting of BCL-2 Family Members during Anticancer Treatment: A Necessary Compromise between Individual Cell and Ecosystemic Responses?

**DOI:** 10.3390/biom10081109

**Published:** 2020-07-25

**Authors:** Sophie Barillé-Nion, Steven Lohard, Philippe P. Juin

**Affiliations:** 1Centre de Recherche en Cancérologie et Immunologie Nantes Angers (CRCINA), INSERMU1232, Université de Nantes, F-44000 Nantes, France; sophie.barille@univ-nantes.fr (S.B.-N.); steven.lohard@nih.gov (S.L.); 2SIRIC ILIAD, 44000 Nantes, France; 3Radiation Oncology Branch, National Cancer Institute, Bethesda, MD 20892, USA; 4Institut de Cancérologie de l’Ouest, 15 Rue André Boquel, 49055 Angers, France

**Keywords:** BH3 mimetics, MOMP, cancer, tumor ecosystem

## Abstract

The imbalance between BCL-2 homologues and pro-death counterparts frequently noted in cancer cells endows them with a cell autonomous survival advantage. To eradicate ectopic cells, inhibitors of these homologues (BH3 mimetics) were developed to trigger, during anticancer treatment, full activation of the canonical mitochondrial apoptotic pathway and related caspases. Despite efficiency in some clinical settings, these compounds do not completely fulfill their initial promise. We herein put forth that a growing body of evidence indicates that mitochondrial integrity, controlled by BCL-2 family proteins, and downstream caspases regulate other cell death modes and influence extracellular signaling by committed cells. Moreover, intercellular communications play a key role in spreading therapeutic response across cancer cell populations and in engaging an immune response. We thus advocate that BH3 mimetics administration would be more efficient in the long term if it did not induce apoptosis in all sensitive cells at the same time, but if it could instead allow (or trigger) death signal production by non-terminally committed dying cell populations. The development of such a trade-off strategy requires to unravel the effects of BH3 mimetics not only on each individual cancer cell but also on homotypic and heterotypic cell interactions in dynamic tumor ecosystems.

## 1. Introduction

Mitochondria are essential for life due to their position at the core of cellular metabolism and respiration. At the same time, they constitute the most actionable intracellular organelle to execute active cell death and ensure cell and tissue homeostasis. BCL-2 family members play a key executioner role by integrating various exogenous or intracellular signals into loss or maintenance of mitochondrial outer membrane (MOM) integrity. Upon overwhelming stress conditions, cells with an altered network of BCL-2 proteins, favoring BAX/BAK pore forming activity engage into a process of complete mitochondrial outer membrane permeabilization (MOMP), leading to massive activation of caspases. The latter proteases cleave many substrates actively preparing cells to die by apoptosis.

Apoptosis is crucial during development and for tissue homeostasis, but is often impaired in cancer. Adaptation to apoptosis is now understood to be a fundamental step not only during tumor natural course (for initiation or progression), but also for tumor resistance to anticancer treatments including chemo- and radiotherapies. Resistance to treatment occurs through pre-existing or acquired mechanisms relying on genetic modification and/or rewiring of intracellular signaling pathways. It is a major factor driving tumor relapse with frequent metastasis and ultimately cancer-related deaths. The contribution of the mitochondrial apoptotic pathway, and of BCL-2 family members that regulate it, to cancer cell resistance justifies the development of agents targeting MOMP. Among these, BH3 mimetics are small molecules developed to inhibit pro-survival proteins (BCL-2, BCL-xL, MCL-1), unleashing BAX/BAK to promote MOMP and apoptosis onset in otherwise aberrantly surviving cancer cells. Their success in treating hematological malignancies opens new therapeutic opportunities for a wider use in oncology. The dose-limiting secondary effects of these molecules nevertheless imply that more efficient therapeutic strategies need to be designed to fully exploit their therapeutic potential. This involves the development of predictors of efficiency in cancer cells, but also better understanding of the cross talks between apoptotic cell death and other modes of cell death whose triggering may lead to biological variations involved in cancer progression. The influence of these cross talks on the tumor microenvironment and the immune system also need to be deciphered. As a matter of fact, BAX/BAK dependent MOMP leads not only to cytochrome-c (cyto-c), SMAC cytosolic translocation and caspase activation to execute cell death, but also trigger viral mimicking inflammation. Moreover, caspases downstream of MOMP also influence proinflammatory signaling which contributes to tumor response or resistance to treatment and shape antitumoral immune response. We herein review how MOMP, and BH3 mimetics may modulate other cell death modalities and intracellular communications in the context of cancer treatment and discuss how it could reshape tumor ecosystem dynamics upon cytotoxic treatment, to better control tumor response and improve survival.

## 2. BCL-2 Family Finely Tunes MOMP and Subsequent Apoptosis in Tumor Cells in Response to Cellular Stress Including Those Induced by Anticancer Treatments

Many studies have reported the contribution of BCL-2 family to tumor initiation, progression, or resistance to therapy [1,2,3]. Alterations in the BCL-2 family include upregulated expression of the prosurvival BCL-2 family proteins BCL-2 (in lymphoma), MCL-1 (whose gene is amplified in 10% of cancers), or BCL-xL (in many chemoresistant cancer cells) or downregulated expression of key apoptotic effectors (for example BIM in Burkitt lymphoma) [4]. This is understood as one common mechanism for cancer cells to increase their anti-apoptotic defense mechanisms, and acquire a selective survival advantage in response to intrinsic oncogenic stress, extrinsic microenvironmental death signaling, and anticancer therapies. Many relapsing tumors are also associated with metastasis, which is the leading cause of cancer-related deaths. Therefore, it is critical to identify the mechanisms of chemoresistance to develop targeted therapies and improve the rate of relapse free survival. Resistance to chemotherapy and poor prognosis is often correlated with enhanced activity of anti-apoptotic BCL-2 homologues that are now therapeutic targets.

Each member of BCL-2 family contains one or more BCL-2 homology (BH) domains (BH1 to BH4), discriminating multi-BH domain proteins (either pro- or anti-apoptotic) from BH3-only (activators or sensitizers) members. The multi-BH domain proteins BAX and BAK are the central pro-apoptotic members that execute MOMP. In addition, BOK an unconventional BAX/BAK-like BCL-2 effector, can also induce MOMP (even in the absence of BAX/BAK), but in contrast to BAX/BAK its activity on MOMP is constitutive and not regulated by BCL-2 family members but by proteasomal degradation [5]. In normal conditions, BAX/BAK are kept in inactive state in the cytosol or in the MOM by the binding of the prosurvival proteins, BCL-xL, BCL-2, or MCL-1 (and BCL2A1, BCL-w). In stress condition, they are activated, by a series of conformational changes that allow them to insert into the MOM where they assemble as pore-forming oligomers favoring the release of apoptogenic proteins (mainly cytochrome-c (cyto-c)) from the intermembrane space (Figure 1). BAX and BAK can be directly activated by binding to the BH3-only direct activators (BID, PUMA, or BID) [6,7,8] or by other proteins such as TP53 [9]. It should be noted, however, that the absence of these activators does not absolutely prevent inhibition of anti-apoptotic BCL-2 proteins from leading to BAX/BAK mediated apoptosis [10]. This implies that other, yet to be characterized modes of BAX/BAK activation (e.g., relying on specific lipid binding) may intervene. Alternatively, in certain instances, antagonism of BCL-2 homologues may suffice to unleash BAX/BAK pore forming activity, by a process fueled by the remarkable self-amplificatory nature of the BAX/BAK activation process [11,12].

BCL-2 homologues prevent initiation and execution of MOMP by binding to BH3-only activators and to activated BAX/BAK molecules, respectively. Both types of interactions occur via the same binding of the hydrophobic face of the BH3 domain on the proapoptotic member into the hydrophobic groove formed by the BH1-BH3 domains in prosurvival proteins. This explains why binding to this interface (in MCL-1 or BCL-xL, respectively) by BH3-only sensitizer proteins NOXA or BAD alleviate BAX or BAK from their apoptotic load and promote their activation. Sensitizers essentially function as competitive inhibitors of interactions between anti-apoptotic members and activators (or active BAX/BAK). Yet, proteins such as BCL-xL may be part of higher-order complexes engaging multiple BH3-only proteins that can be allosterically regulated by BH3-sensitizer in a non-competitive mode [13]. Thus, MOMP onset is determined by the relative abundance and binding affinities between pro- and anti-apoptotic proteins but also by the mode of assembly of the latter and, as discussed below, by their subcellular localization. Despite apparent redundancy, these interactions are selective as they differ by their affinity (Figure 1). For example, BAD preferentially binds to BCL-2 and BCL-xL whereas NOXA preferentially binds to MCL-1 so that these sensitizers act as endogenous inhibitors of subsets of complementary anti-apoptotic proteins. BID preferentially activates BAK, and BIM preferentially binds BAX implying that cancer cells lacking BAK are relatively resistant to agents that require BID activation for maximal induction of apoptosis [7,14]. Even though BAX and BAK are considered redundant since only their combined loss leads to apoptosis resistance [15], BAX differs from BAK by the opposing effect of mitochondrial VDAC2 on their activity [16,17]. Different mitochondrial proteinaceous receptors are indeed involved in the interaction of BAX and BAK with mitochondrial outer membranes [18,19]. Importantly, expression of BCL-2 proteins varies in normal tissues where apoptotic priming is developmentally regulated [20].

The biochemical consequences of MOMP driven by BAX and BAK is in great part due to the release of soluble proteins from the mitochondrial intermembrane space, such as, as mentioned above, cyto-c, but also SMAC/DIABLO or Endonuclease G. Cyto-c in addition to its contribution to the electron transport respiratory chain in the intermembrane space, allows the formation of the apoptosome together with the adaptator molecule APAF-1 and the downstream activation of the caspase cascade when in the cytosol. The apoptotic executioner caspases are responsible for the cleavage of several hundreds of substrates and the characteristic morphological changes of apoptosis that include membrane blebbing, cell shrinkage, the formation of ‘‘apoptotic bodies,’’ and chromosomal DNA fragmentation.

Many tissue specific transcriptional regulations and protein degradation processes, by the ubiquitin-proteasome system in particular, govern BCL-2 family proteins’ levels in cells [21]. Cancer cells often harbor increased levels of pro-apoptotic BH3 proteins due to chronic environmental stressors, but they maintain their survival by expressing enough prosurvival proteins to buffer these proapoptotic proteins. Genotoxic chemotherapies frequently upregulate BAX, PUMA, or NOXA genes’ expression through transcriptional activation of p53 that changes its function from repairer to killer upon deleterious DNA damage [22]. BCL-2 proteins undergo cell cycle-dependent modifications such as mitotic BCL-xL phosphorylation on serine 62 that acutely disables BCL-xL/BAX interaction promoting mitotic apoptosis [23]. Most BCL-2 homologues exhibit a C-terminal transmembrane domain that defines them as tail-anchored membrane proteins (TAMPs) [21]. As a result, intracellular membranes play an active role in BCL-2 family interactions by regulating their local relative abundance and enforcing the binding of anti-apoptotic proteins to pro-apoptotic counterparts, so that BCL-2 homologues shuttling to and from mitochondrial membranes is critical for the outcome of protein–protein interactions [24,25]. Other unrelated interacting proteins, such as E2F1 modulate the BAK-sequestrating prosurvival activity of BCL-xL by interfering with its intracellular motility [26].

How quantity, stability, activity, and subcellular localization of BCL-2 family proteins tightly control MOMP onset in connection with the cellular context and regulation of BCL-2 proteins on the mitochondrial surface, is still subject to intensive research [7,27,28]. Predisposition to MOMP triggering is multifactorial and it is difficult to univocally assign to it to a specific gene expression signature or proteomic profile prior stimulation. Letai and colleagues developed a technique called BH3 profiling to evaluate propensity to MOMP induction at cellular level. It relies on quantifying MOMP (measuring mitochondrial potential collapse or release of cyto-c as a readout) in cells exposed to synthetic 20-mer BH3 peptides, mimicking the proapoptotic function of BH3-only proteins. These latter are chosen for their selectivity and used to reveal the cell dependence to BCL-2 prosurvival proteins [29,30]. Cancer cell sensitivity to this assay prior treatment is predictive of a clinical chemotherapeutic response [31]. This has led to the notion that cancer cell priming for mitochondrial apoptosis is critical: a lack of priming in tumors is sufficient for intrinsic chemoresistance while a primed state is necessary for chemosensitivity. Consistently, enhancement of priming, detected by dynamic BH3 profiling, is an early mark of therapeutic efficiency [32].

The fact that cancer cells strongly rely on BCL-2 homologues for survival maintenance is particularly blatant in hematological malignancies [20] but single or co-dependencies towards BCL-2, BCL-xL, or MCL-1 survival proteins were also identified in 50% of solid tumor cell lines [33]. This leads to the hypothesis that manipulating MOMP using drugs that target the BH3 binding activity of BCL-2 homologues to promote apoptotic caspase activity, may improve cancer treatment. Importantly, enhancement of BCL-2 homologues activity facilitates cancer cells to escape the anticancer cytotoxic effect but at the same time often render them tenuously dependent on this continuous block to survive [33]. This notion paved the way for the development of BH3 mimetics as pro-apoptotic anticancer agents used as single agents or as chemosensitizers.

## 3. BH3 Mimetics as Potent Cancer Cell Killers and Antagonists of Chemoresistance

Great efforts were dedicated to the development of molecules that by inhibiting prosurvival protein functions, promote apoptosis in cancer cells. Small molecules called BH3 mimetics, that bind into the BH3 binding pocket of the prosurvival proteins BCL-2, BCL-xL, or MCL-1 (as BH3 domains of the proapoptotic ones do), favor mitochondrial apoptosis in cancer cells and have already useful clinical applications. In freeing the pro-apoptotic proteins sequestered by the prosurvival proteins, these molecules increase a proapoptotic load on BAX and BAK and promote subsequent MOMP in cells that are highly dependent on these proteins for their survival (Figure 1). However, antiapoptotic adaptation in response to therapies varies between cancer types, from tumor to tumor and maybe from one cancer cell to another. Determining its molecular support to precisely tackle it, and conceiving BH3 mimetics as tools for ecologic therapy as discussed below, is thus of major importance for their efficient use.

The first BH3 mimetic ABT-737 was discovered by Oltersdorf et al. in 2005 using nuclear magnetic resonance (NMR)-based screening of a chemical library to identify small molecules [34]. ABT-737 binds to the hydrophobic BH3-binding groove of BCL-xL recapitulating BAD-BH3 domain, with additional on target effect on BCL-2 and BCL-W. ABT-737 exhibited single-agent-mechanism-based killing of cells from lymphoma and small-cell lung carcinoma, and enhanced the effects of death signals, displaying synergistic cytotoxicity with chemotherapeutics and radiation in various cancer cell lines. The proof of ABT-737 or its orally bioavailable derivative ABT-263 improved survival, causing regression of established tumors in many preclinical studies, launched the development of BH3 mimetics for use in oncology [35].

Exploiting the subtle differences between the binding interfaces of BCL-2 versus BCL-xL led to the identification of the BCL-2 specific (selective) BH3 mimetic ABT-199 or venetoclax. In 2016, US Food and Drug Administration (FDA) approved venetoclax for treating chromosomal 17p-deleted chronic lymphocytic leukemia (CLL) used as single agent [36,37]. Remarkable response in patients with CLL even those who have failed standard chemo-immunotherapy, has been achieved. Patients who achieve a deep response (i.e., with a negative minimal residual disease) can stop their treatment and still sustain remission. Predictive biomarkers are however needed to prospectively identify which patients will benefit from BH3 mimetic-based therapies, since high level of BCL-2 expression is required but not sufficient to be predictive. Genomic instability (complex karyotype and resistance to fludarabine therapy) is so far the most powerful predictor of failure of venetoclax monotherapy in CLL. In contrast to CLL, more heterogeneous results were obtained in other blood cancers such as B-cell malignancies [38,39]. Although single agent clinical activity with venetoclax has been modest in acute myeloid leukemias (AML), clinical responses are increased in combination with either hypomethylating agents or low-dose cytarabine. Based on its low toxicity, venetoclax has thus received FDA approval in combination with hypomethylating agents for the treatment of newly diagnosed elderly patients ineligible for intensive chemotherapy with AML. Many studies reported synergistic effects obtained combining various chemotherapies with BH3 mimetics in preclinical models and clinical trials are still ongoing for complete evaluation.

Targeting BCL-2 is most advanced clinically and has shown great potential in hematological malignancy but less efficacy in solid cancers. While hematopoietic malignancies appear addicted to a single pro-survival protein, the survival of cancer cells in carcinoma is often safeguarded by multiple pro-survival BCL-2 family proteins [40]. For instance, BCL-2 inhibitors synergized with tamoxifen or chemotherapy and decrease tumor growth in preclinical models with highly BCL-2 expressing ER-positive or triple negative breast tumors [41,42]. However, as cancer cells may be dependent on other anti-apoptotic proteins due to their heterogeneity and phenotype/genotype plasticity, BH3-mimetics targeting the other prosurvival proteins BCL-xL or MCL-1 are also undergoing clinical investigation [43].

Selective BCL-xL inhibition could be especially useful to treat tumors that often regulate BCL-xL as a mechanism of adaptation [44]. Importantly, several studies demonstrate that BCL-xL is required for cell survival in physiological or chemo-induced specific cellular contexts. We and others have shown that mitotic cell survival upon antimitotic treatment mainly relies on BCL-xL revealing the opportunity to use BCL-xL inhibitors in combination with antimitotics to improve their cytotoxic effect [23,45,46]. BCL-xL also sustains viability during drug-induced polyploidization upon treatment by an auroraB inhibitor [47]. Across solid tumor cell lines, BCL-xL dependence was significantly and positively correlated with a mesenchymal signature compared to cancer cells harboring an epithelial phenotype. As a consequence, (possibly chemo-induced) epithelio-mesenchymal transition (EMT) in relation to ER stress-induced PERK signaling activation might increase dependence on BCL-xL and render necessary the use of specific inhibitors thereof [33,48,49]. Mechanistically, increased NOXA expression, observed upon antimitotic treatment [46,50,51], EMT [33] or ER stress [52], is expected to modify dependence of cancer cells on BCL-2 family proteins, shifting the burden to support viability from MCL-1 (buffered by NOXA) to BCL-xL exclusively. Dynamic NOXA expression may thus help predict BCL-xL inhibitors efficiency. Using a high-throughput screen to discover a new series of small molecules targeting BCL-xL and their structure-guided development, Lessene and colleagues identified the first BH3 mimetic selectively targeting BCL-xL WEHI-539 [53] and additional BCL-xL-selective inhibitors exhibiting oral bioavailability were further characterized [54]. These compounds are responsible for dose-limiting thrombopenia due to their BCL-xL on target effect in platelets [55], but they have the potential to enhance the efficacy of docetaxel in a range of solid tumors in various preclinical models and now undergo clinical trials in oncology.

Targeting MCL-1 by small molecules was more challenging because of its more complex structure, and several MCL-1 inhibitors have been produced using different strategies showing modest selectivity. However, the novel molecule S63845 exhibits high selectivity for MCL-1 over BCL-2 or BCL-xL and shows promising results with good tolerance in preclinical studies [56]. MCL-1 may represent a potent target to treat breast tumors in particular, since Her2-amplified or chemotherapy-treated TNBC breast cancers are probably prone to MCL-1 dependence [57,58].

Overall, combining BH3 mimetics with chemotherapy is under intense investigation, in particular in solid cancers that exhibit more heterogeneous and evolutive survival dependencies towards BCL-2 family proteins (Table 1). Targeting BCL-xL/BCL-2 with the first available dual inhibitors ABT-737 and navitoclax, exhibit potent synergy with several chemotherapeutics in difficult-to-treat tumors such as triple negative breast cancers or non-small cell lung carcinoma. BH3 mimetics inhibiting BCL-xL greatly enhanced antimitotic efficacy in exploiting the BCL-xL dependence resulting from NOXA induction by antimitotics [50]. BH3 mimetics also synergized with other on-target therapies such as inhibitors of oncogenic kinases that often induce upregulation of pro-apoptotic BH3-only proteins BIM, PUMA, or BMF and shifts between BCL-2 prosurvival dependence in cancer cells. Combination of MCL-1 inhibitors with inhibitors of EGFR, MEK, or B-RAF showed potent antitumor activity in various preclinical models of cancer [59,60].

## 4. Toxicity and Resistance-Limitations of BH3 Mimetics Efficacy in Clinical and Preclinical Settings

The potential for BH3 mimetics to selectively kill cancer cells over normal cells is based on the concept of their higher apoptotic priming due to apoptotic oncogenic signaling and/or microenvironmental apoptotic pressure [7,20]. In fact, mitochondria from adult somatic tissues resist pro-apoptotic signaling. This resistance is higher than in mitochondria from younger tissues, as a result of a decrease of c-Myc driven expression of the mitochondrial apoptotic machinery over age [20]. Even though these features open therapeutic windows for BH3 mimetics, these nevertheless have on-target toxicities, such as thrombopenia upon BCL-xL inhibition [55], or neutropenia upon BCL-2 inhibition [54]. Several clinical trials using ABT-263 indicate that its safety on platelets in particular, can be controlled using appropriate dosing [61,62]. Interestingly, the ABT-263-derived new compound based on proteolysis-targeting chimera (PROTAC) technology targets BCL-xL for degradation by VHL-E3 ligase and spares platelet-expressed BCL-xL as these cells minimally express VHL [63]. Another PROTAC recruiting Inhibitors of Apoptosis proteins (IAP) also achieved BCL-xL degradation in cancer cell lines opening the avenue of selective BCL-xL degraders based on specific cellular E3 ligase activities [64].

Despite complete response rates of up to 50% in CLL treatment by venetoclax, secondary resistance is the most frequent cause of treatment failure. Resistance mechanisms observed in CLL patients treated with venetoclax include the acquisition of BCL-2 mutations such as the Gly101Val mutation that reduce venetoclax binding to BCL-2, or compensatory overexpression of other pro-survival proteins BCL-xL and MCL-1 [65,66,67,68,69,70]. Complex clonal shifts including mitochondrial metabolic reprogramming with altered expression of the AMPK signaling pathway, or mutations in BTG1 and aberrations of CDKN2A/B have been observed in venetoclax-resistant CLL [66,71]. In AML the mitochondrial chaperonin CLPB that maintains cristae structure via its interaction with the cristae-shaping protein OPA1, also contribute to venetoclax resistance [67]. These compensatory processes arising from pre-existing/acquired selection or adaptive activation, alone or associated, define disease progression after BH3 mimetic-based therapies as an emerging therapeutic challenge, as previously observed with tyrosine-kinase inhibitors used in CML. As intrinsic resistance to BH3 mimetics can be mediated by untargeted prosurvival BCL-2 family members including BCL-w or BCL2A1 [33], blocking all antiapoptotic proteins by combining several BH3 mimetics may be theoretically envisioned since, as noted above, apoptotic resistance has been observed in many vital organs in adults [20].

In addition to the intrinsic resistance to tumor cells, a growing body of evidence indicates that support of tumor cell survival by cancer-associated fibroblasts (CAFs, a main component of the cellular microenvironment) critically favors cancer progression and regrowth post therapy [72]. We indeed observed that CAF modulate BCL-2 dependence in luminal breast cancer cells via IL-6 signaling, shifting from a BCL-2 to a MCL-1 survival dependence [73]. In addition, since CAFs mitochondrial integrity relies on MCL-1, MCL-1 inhibition might be particularly relevant to target them. This puts forth the notion that BH3 mimetics may be viewed as the basis of an ecologic anticancer treatment. Along this line, the fact that inhibitors of BCL-xL may act as senolytic drugs is relevant [74] since, chemo- and radiotherapies tend to induce a senescent phenotype in microenvironmental cells [75] (and references therein) that influences treatment outcome. BH3 mimetics may improve therapy efficiency by eliminating senescent cells, thereby influencing the response of tumor ecosystem as a whole.

## 5. Limitations of BH3 Mimetics Efficacy: Fractional, Adaptive or Incomplete Biologic Responses

Failures of BH3 mimetics may rely on incomplete MOMP, on an unprimed cellular state or on survival addiction of cancer cells to other/several antiapoptotic proteins.

MOMP is often rapid and complete, enrolling the majority of mitochondria in a given cell after apoptotic stress. Under these circumstances extensive MOMP appears as an all-or-nothing and as a point of no return [76], possibly due to the existence of amplificatory mechanisms of BAX/BAK activation and/or of ROS production that propagate MOMP in cells. In addition, intrinsic cancer cell-to-cell variability can lead to fractional killing upon apoptotic stress due to fluctuations in proteins levels whose temporal dynamic is fundamental to the onset of apoptosis [77,78]. We indeed reported that treatment of BCL-xL-dependent cancer cell lines with BH3 mimetics targeting BCL-xL, systematically spares individual cells with the highest levels of this protein suggesting BH3 mimetic-resistant residual binding between BCL-xL and proapoptotic proteins [24]. Our results also suggest that when stably localized at membranes instead of shuttling between mitochondria and cytosol, BCL-xL prosurvival activity is enhanced by selectively enforcing its binding to BH3 activators BIM and PUMA, making refractory the pro-apoptotic effects of ABT-737.

In some instances, at the level of a single cell, mitochondria may in fact undergo permeabilization in a less coordinated manner as usually understood, and this might contribute to resistance. An incomplete MOMP allowing cell survival was evidenced, with intact mitochondria repopulating a viable mitochondrial network [79]. Incomplete, non-lethal MOMP could be detected when downstream caspase activity was blocked while GAPDH levels were sufficient to ensure ATP production and mitophagy was going on [80]. In the same line, a mild apoptotic stress may engage a low level of MOMP or a low fraction of mitochondria undergoing MOMP in a cell (namely minority MOMP) that may trigger a caspase activity too low to commit cell death. In cancer cells, this minority MOMP may contribute to long term resistance and aggressivity. Actually, by triggering the caspase-activated DNAse (CAD), incomplete MOMP can result in genomic instability and in resistance to anticancer therapies [81,82,83]. When MOMP is incomplete, refractory mitochondria may be characterized by higher levels of BCL-2 homologues at their surface [79].

Insufficient BH3-only proteins in cells also results in resistance to MOMP and cyto-c release after mitochondria exposure to sensitizer BH3 peptides [84]. Resistance to BH3 mimetics characterizes this cellular state that is called unprimed for apoptosis. This apoptosis refractory state is observed in various normal adult tissues offering a therapeutic index in cancer treatments [20] but also characterizes apoptosis resistant cancers that have to be identified to adapt therapies.

At cellular level, resistance to BH3 mimetics may also be influenced by dynamic changes in apoptotic signaling induced by anticancer treatments. Montero and colleagues interrogated the dependence of cancer cells on the BCL-2 family proteins after targeted cancer therapies and observed that multiple oncogenic driver-targeting therapies, including BRAF or EGFR inhibitors, eventually downregulate NOXA expression because the MAPK-dependent pathway favors NOXA mRNA stabilization. This generates a dependence on MCL-1 in treated cancer cells, that can be therapeutically overcome by sequential inhibition of BRAF and MCL-1 in preclinical studies [85]. The efficacy of this combination could even be enhanced by prior transient exposure to BCL-xL inhibitors which promote the binding of pro-apoptotic proteins (in particular BIM) from BCL-xL to MCL-1 and by this way augment the tumor MCL-1 dependency [60]. Of note, experimental approaches and mathematical modelling hint that chemotherapy may induce a transitory tolerant cell state associated with activation of various kinases and suppression of apoptosis [86]. This metastable phenotypic state may converge towards metabolic plasticity that confers a survival advantage to cancer cells [87]. This can be reversed by kinase inhibitors but whether BH3 mimetics exposure could interfere with such processes remains to be determined.

Another limitation to the use of BH3 mimetics is that they may inhibit neither fully the canonical function of their targets nor their non-canonical functions. Prosurvival BCL-2 proteins may interact with their pro-apoptotic counterparts through their BH3-binding site but also through other interfaces, as described by Andrews and colleagues for interactions between BIM and either BCL-2 or BCL-xL, that engage into ABT-737 resistant complexes [88]. Antiapoptotic proteins encompass other active BH domains such as the BH4 domain in BCL-xL that contributes to its protumorigenic activity involving RAS or VDAC1 [89,90]. How BH3 mimetics interfere with these BH3 independent activities is not known.

With the initial view of MOMP as an action switch button in the apoptotic process, BH3 mimetics offered the promise of triggering massive and complete apoptosis in cancer cell populations. However, as mentioned above, extensive exploration of BH3 mimetics effects and resistant clinical cases underline that their action is disappointing on that aspect. Indeed, these compounds induce fractional killing at best (like many other cytotoxic agents), occasionally sparing mitochondria subsets and possibly failing to induce full caspase activation. This may cast doubt on their usefulness as anticancer agents in the long term in solid tumors. As discussed below, however, the use of BH3 mimetics as laboratory tools to explore the biological effects of mitochondrial integrity alterations has, at the same time, helped redefine the biological role of MOMP and highlight the abundant intracellular crosstalks between distinct cell death pathways. In part because it corresponds to a rupture of the symbiotic relationship between mitochondria and the host cell, MOMP is involved in cell death modes that are more inflammatory than apoptosis stricto sensu and caspases regulate this process. Thus, in addition to target cancer cells, BH3 mimetics may be useful to recruit and amplify the effects of anti-tumoral immunity.

## 6. MOMP Contributes to an Inflammatory Signaling and to Immunogenic Cell Death Modes

Mitochondrial apoptosis is classically regarded as immunologically silent. This feature is achieved by limited plasma membrane breakdown, selective release of cellular contents, and display of ‘‘find me’’ and ‘‘eat me” signals by apoptotic cells to trigger the rapid phagocytic clearance of apoptotic bodies. Chekeni and colleagues showed that apoptotic cells are not passive during this process of removal; they actively maintain certain metabolic pathways and only selected metabolites are released through caspase-mediated opening of pannexin 1 channel in the plasma membrane. These function as tissue messengers triggering in neighboring cells proliferation, wound healing, and suppression of inflammation [91]. However, when uptake of apoptotic corpses by macrophages (efferocytosis) is defective, apoptosis induction may lead to secondary necrosis, in which rupture of the plasma membrane triggers release of Damage Associated Molecular Patterns (DAMPs) that stimulate inflammatory and immunogenic reactions [92]. Release of DAMPs and metabolites is a key determinant of the consequences of cell death. Dying cells release constitutive DAMPs (ATP or HMGB1 for example), but also inducible ones that rely on maintained RNA transcription and protein translation [93]. Importantly, the nature of the inducible DAMPs depends on the type of cell and on the death pathways that are engaged but in general NF-kB pathway activation and type I-IFN production are essential contributors [94]. MOMP is connected to cell death modalities that trigger these pathways (Figure 2).

Paradoxically, with regard to its major role in apoptotic cell death, MOMP is inflammatory. In 2014, two studies (using BH3 mimetics) showed that during BAX-BAK dependent MOMP, mitochondrial DNA (mtDNA) is released in the cytosol where it activates the cytosolic DNA sensor pathway cGAS/STING to promote, as does viral infection, an inflammatory reaction with type I IFN expression [95,96]. Mechanistically, recent studies described large pores in the MOM through which the inner mitochondrial membrane carrying with it mitochondrial matrix components, including the mitochondrial genome, herniate before rupturing into the cytosol. This inner membrane permeabilization occurs after activation of BAK/BAX and cyto-c loss but independently from caspase activity and it enables inflammatory signaling [97,98]. These studies centered on MOMP evoke a more general feature: mitochondrial DNA, somehow in agreement with its bacterial origin, is sensed not only as an extracellular, but also as an intracellular DAMP when in the cytosol. Its relatively low methylation status, and its propensity to oxidative damage, compared to nuclear DNA, seems to be critical [99]. Numerous studies have now established that mtDNA can stimulate many Pattern Recognition Receptors (PPRs), including cGAS as mentioned above but also TLR9 and inflammasomes (reviewed in [100]) and how all these can be triggered upon MOMP remains to be elucidated.

In the case of cGAS activating (viral mimicking) MOMP, induction of type I interferon (type I-IFN) pathway is blunted by downstream caspases and only unmasked when caspases are inhibited or genetically ablated (Figure 3). This negative regulation is mechanistically understood as due to inactivating cleavage of cGAS and IRF3 [101]. A notion that arises from this is that, during cell death induction by MOMP, apoptotic caspases do not necessarily determine cell fate, but rather death modalities. It is notable that apoptotic caspases are ultimately dispensable for cell death and clearance of apoptotic cells in vivo (as observed in hematopoietic system) in contrast to BAK and BAX [96,102]. By analyzing the fate of cells undergoing (BH3 mimetic induced-) MOMP in the absence of downstream caspase activation, Tait and colleagues established that such cells die through a necroptotic process [103]. This does not necessarily involve mtDNA release but relies on decreased IAPs expression following MOMP (possibly via the release of SMAC-like molecules) leading to NF-κB-dependent TNF synthesis. In this setting, MOMP-induced necroptosis is not only unmasked by caspase inhibition but made possible by it. Regulated necroptosis is indeed a caspase-inhibited, TNFR-induced form of cell death relying on Receptor-Interacting Protein Kinase 1 (RIPK1) and RIPK3 activation. It is inflammatory in particular because it is accompanied with NF-κB pathway activation and type I-IFN production [104]. TNFR1 initiates distinct apoptotic or necroptotic as well as NF-κB activation depending on the recruitment and engagement of regulatory components in macromolecular complexes including caspase-8 and cIAP1/2 in addition to RIPK1/3. When caspase-8 is inhibited during TNFR1 stimulation by TNF, RIPK1 no longer cleaved by caspase-8 activates RIPK3, allowing phosphorylation and oligomerization of the forming pores MLKL in the plasma membrane (Figure 2). Importantly, dying cells can release DAMPs resulting from NF-kB pathway activation and type I-IFN production [93]. cIAP1/2 prevent TNFR1-mediated necroptosis by constitutively ubiquitinating RIPK1 leading to its degradation. Inhibition of cIAP1/2 (using SMAC mimetics or during MOMP) and of caspase-8, drive TNFR1 signaling toward necroptosis and this can promote tumor reduction (e.g., in leukemia preclinical models) [105,106]. A molecular link between regulators of MOMP and necroptosis was established by [107] who showed that PUMA is transcriptionally upregulated by NF-κB upon necroptosis initiation, and that it amplifies necroptosis through PUMA-mediated signaling back to RIPK1 and MLKL. The interplay between MOMP and caspase activation is critical for the systemic outcome of cell death induction: triggering MOMP-induced TNF (by BH3 mimetics), in combination with mtDNA release and cGAS/STING-dependent activation of type I-IFN pathway, exhibits enhanced antitumor activity when caspases are inhibited. This gain of effect relies on inflammation-induced recruitment of macrophage and cytotoxic T-cell infiltration [103]. It should be noted that RIPK3 is often epigenetically silenced in cancer cells so that these cells may neither readily undergo MOMP-induced necroptosis nor produce an inflammatory secretome [108] unless additional epigenetic drugs are used [109].

The antagonistic effects of MOMP and of downstream caspases on the anti-tumoral secretory phenotype of cells committed to viral mimicry and/or necroptosis advocates that caspase inhibition may be beneficial for anticancer treatment and that a contrario, caspase activation by apoptosis inducing agents (which BH3 mimetics are by design) is not. Previous studies have already reported that caspase inhibitors can have antitumor effects [110]. Most recently, the pan-caspase inhibitor emricasan (IDN-6556) has been evaluated in phase 2 clinical trials in patients suffering from non-malignant hepatopathies and it overall failed to provide proof-of-concept support for caspase inhibition as a treatment for non-alcoholic steatohepatitis (NASH) and cirrhosis patients [111,112,113].

Pyroptosis is a form of cell death most commonly occurring as an innate immune response, depending on the cleavage of caspase-1 by inflammasome complexes leading to the activating cleavage of the proinflammatory cytokine IL-1β (a pleiotropic mediator of inflammation) and the cytosolic protein gasdermin D (GSDMD) promoting its capacity to form membrane pores leading to cell lysis [114,115]. Inflammasome-activated caspase-1-dependent pyroptosis damages mitochondria and promotes mitochondrial permeability transition dependent-release of cyto-c, which evokes MOMP even though a role for BAX and BAK remains to be established [116]. It is also notable that GSDMD is specifically enriched in mitochondrial membrane where it triggers mitochondrial injury through ROS generation and downstream NLPR3 inflammasome activation or pyroptosis [117] (Figure 2). Reciprocally, MOMP can promote pyroptosis as another gasdermin (GSDME/DFNA5) can be activated by caspase-3 cleavage [118,119,120] (Figure 3). Expression levels of gasdermins may thus determine whether MOMP triggers canonical apoptosis or a pyptosis-like process. As for necroptosis, this may be more relevant to the response of non-malignant cells of the tumor microenvironment than to that of malignant ones, as these tend to exhibit no expression of GSDME [121]. Another scenario where MOMP and caspases cooperate to promote inflammation is that described by Vince et al. [92] in macrophages. Caspases 3 and 7, activated downstream of BH3 mimetic-induced MOMP, promote in these cells maturation and production of IL-1β by caspase-8 activation and, in parallel by NLRP3 inflammasome, possibly recruited by caspase dependent potassium efflux. Intriguingly, exposure of myeloid and epithelial cells to IL-1β leads to release of mtDNA into the cytosol and subsequent activation of the cGAS-STING pathway [122]. This underscores the complexity of the relationship between MOMP activated caspases and inflammation: a wave of inflammation may propagate, initiated by IL-1β producing cells if they activate caspases, but relayed by IL-1β recipient cells only if these do not activate caspases above a threshold that would inactivate cGAS/STING [101].

These ambiguous connections raise the question of whether caspase activity in a whole tumor is linked in any way to its immune status. A universal role for apoptotic caspase activity in antagonizing inflammatory processes would predict that high levels of active caspases should negatively correlate with inflammation markers and immune infiltration. Exploration of public databases comprising matched proteomic and RNA expression data indicate that this is not always observed (personal computation based on data sets from the Cancer Genome Atlas 2012 [123]). It is intriguing to note that, instead, in breast cancers, ovarian serous cystadenocarcinoma, colorectal adenocarcinoma, or skin cutaneous melanoma, active caspase 7 levels (measured by Reverse Phase Protein Array) positively correlate with numerous molecular markers of cytotoxic immune cell activation (including STAT1, Interferon type II, or granzyme genes) and immune checkpoints (including CTLA4, CD274, or PDCD1), suggesting tonic immune reactions in caspase 7 high tumors. Arguably, identification of the type(s) of cell that predominantly express active caspase 7 in such tumors (e.g., malignant cells or immune cells themselves) is mandatory to characterize whether, and which, causal links might underlie these positive correlations. In all cases, they imply that activity of caspases downstream of MOMP in tumor ecosystems does not necessarily suppress cytotoxic immune infiltration and that it is not incompatible with immune checkpoint inhibition strategies.

Notably, cell death mode in response to MOMP may not only be determined by downstream caspase activation but also by additional mitochondrial modifications. During MOMP, progressive mitochondrial dysfunction linked to electron transport chain decoupling from ATP synthesis, inducing ATP loss and ROS accumulation, mediates a metabolic catastrophe that contributes to cell demise. This underlies the success of ROS-targeting therapeutic strategies (such as ionizing radiation or oxidant-producing chemotherapeutics) in the treatment of cancers. ROS foster the mitochondrial permeability transition linking mitochondrial metabolic dysfunction to apoptosis [124]. Underscoring the role of “mitochondrial infrastructure plasticity” in finely regulating cell fate, opening of mPTP, when transient, has been proposed to initiate extramitochondrial adaptive responses [125]. Likewise, mitochondrial cristae remodeling, is essential both for basal mitochondrial dynamics and for full MOMP [126].

MOMP may also support ferroptosis which is characterized by iron-dependent accumulation of lipid hydroperoxides to lethal levels in close relation with ROS accumulation, that occurs through progressive mitochondria dysfunction. In some circumstances, both cell death pathways ferroptosis and autophagy-dependent cell death may also crosstalk to apoptosis through ROS production related to mitochondria dysfunction or through Beclin interaction with BCL-2 antiapoptotic proteins, respectively (Figure 2). How these other MOMP connected cell death modes influence inflammation and immune responses remains to be established.

## 7. Cell Death at Population Levels: Intercellular Communications in Tumors during Treatment

With the almost systematical, yet pleiotropic, influence of mitochondrial apoptotic stress on inflammation, chemosensitivity, immune response, and tumor progression following (conventional or BH3 mimetics) therapy, arguably relies on how treatments remodel the tumor microenvironment and adaptative immunity. In particular their influence on intercellular communications between malignant clones and non-malignant cells is critical: committed or dying cells can generate active signals that drive tumor response to anticancer treatments.

It is established that exposure to chemotherapeutic agents can change the types and abundance of components in the tumor secretome. For example, IL-6 and IL-8 are frequently induced, and their expression strongly correlates with tumor recurrence and poor responsiveness to therapy in various cancers [127]. They protect many cancer cells from chemotherapy through various mechanisms including induction of antiapoptotic proteins BCL-2, BCL-xL, and XIAP [128,129]. Such factors can be produced by the tumor environment: we indeed recently reported that IL-6 produced by CAF protect luminal breast cancer cells from apoptosis by increasing and stabilizing MCL-1 expression in these cells [73]. Chemo- or radiotherapies can initiate a senescent program in cells, leading to the production of a pro-inflammatory secretome as a part of senescent-associated secretome phenotype or SASP. It coincides with an accumulation of dysfunctional mitochondria in relation to nuclear chromatin fragments in the cytosol [130,131]. This can promote local and systemic inflammation that causes or exacerbates many side effects of the treatments [132]. Therapy-induced tumor secretome may promote tumor recurrence by enabling cancer cell survival but also the expansion of cancer stem cells, or by preventing antitumor immunity (reviewed in [127]). Other chemotherapy-induced cellular stresses were recently described. These include ER stress that can lead through intrinsic TRAIL receptors ligand-independent signaling, to the production of inflammatory cytokines. The general view is that many “stress-associated molecular patterns” (SAMPs) contribute to chemotherapeutic stress-induced inflammation [133]. These chemo-stress phenotypes induced in cancer cells may depend on the nature of the anticancer drug and its cytotoxic activity. We indeed recently reported that proliferating breast cancer cells produce a TNFα and type I-IFN containing secretome upon antimitotic treatment. This relies on the activation of cGAS/STING pathway by antimitotic-induced micronuclei and leads to the spreading of an apoptotic predisposed phenotype in neighboring non proliferating cancer cells [50] (Figure 4). In the same way, cells committed to die can continue to signal to living cells in the surrounding tissue by producing cytokines during necroptosis as well as tumor immune response [108], or metabolites during apoptosis through pannexin-1 as mentioned above [134].

In addition to soluble messengers, committed cells also provide extracellular vesicles that can transport a variety of cellular contents opening an intercellular communication that emerges as an important mechanism in the development of chemoresistance [135] (Figure 4). As an example, chemo-treated breast cancer cells release DNA containing exosomes that activate dendritic cells via STING signaling [136]. Following MOMP cytosolic mtDNA can move from cell to cell, across a population of cancer cells and engage the cGAS/STING signaling pathway and downstream type I-IFN on neighboring immune cells [137]. Entire mitochondria can migrate from the tumor microenvironment to cancer cells via tunneling nanotubes between endothelial cells and cancer cells or in reverse from cancer cells to tumor microenvironment [138,139]. This horizontal mitochondrial transfer may also take place between healthy cells and early apoptotic cells, rescuing them from cell death onset in an in vitro model [140]. To the best of our knowledge, whether this can happen upon BH3 mimetics has not been determined. Finally, apoptotic debris produced by drug-treated tumor cells readily stimulate macrophages to secrete proinflammatory cytokines and bioactive lipids creating a pro-tumorigenic microenvironment [141]. The phosphatidylserine (PS) exposed on the outer leaflet of the plasma membrane strongly contributes to this effect.

Altogether, these observations indicate that tumor cells, including those dying upon anticancer treatments, intensively communicate with the tumor environment through many routes and that these communications critically shape the global therapeutic response. The numerous connections between MOMP and “immunogenic” cell death modes imply that BH3 mimetics, even if they target limited cell subsets, may nevertheless be useful if they exploit the anti-tumoral effects of these communications (spreading therapeutic response) and/or if they can target cells that produce pro-tumoral signals. Our recent data argue that efficient strategies will have to find a compromise between the biochemical pathways leading to the production of death signals and the own vulnerabilities of cells elaborating them. Indeed, we found that cGAS active cancer cells following anti-mitotic treatments produce a TNF/type I-IFN pro-apoptotic secretome that induces NOXA expression in neighboring cells, rendering them dependent on BCL-xL (Figure 4). These donor cells are themselves sensitized to BCL-xL inhibition, by a mechanism that is unclear but that evokes the fact that STING activation in immune cells induces NOXA and PUMA gene transcription via the coordinated action of IRF3 and p53 (underlying the efficiency of STING agonists in lymphoma preclinical models) [142]. Therefore, treatment of donor cells with BH3 mimetics prevents them from mounting a pro-apoptotic secretome and might prevent spreading of death signaling across population. As a consequence, sequential schedules in combination protocols based on delayed BH3 mimetics administration, were more efficient to limit tumor progression than the same synchronous combination, which may underestimate BH3 mimetic anticancer potential [50]. The inflammatory nature of the intercellular communications underscores that the dose and time schedule of drug administration need to be conceived on the basis of their effects not only on cancer cell populations but also on host response. However, type I-IFN in addition to enhance mitochondrial apoptotic priming and recruit the immune system, yet may promote immune checkpoint expression in the long term [143]. Therefore, destruction of type I-IFN exposed cancer cells is eventually required, justifying a timely use of BCL-xL inhibitors (Table 1). Another key determinant is the impact of BH3 mimetics on antitumoral immune response. In a preclinical model of MYC-driven breast cancer, PD-1+/CD8+ T-cell infiltration was observed after a proapoptotic targeted therapy including venetoclax, that contributed to obtain durable antitumor responses [144]. This suggests that BCL-2 inhibition may leave intact some anti-tumoral immune cells but further dedicated analysis exploring the effects of the whole BH3 mimetic arsenal on human immunocompetent models is required.

## 8. Concluding Remarks

Progress made in the past decades in understanding how the BCL-2 family of proteins controls mitochondrial integrity and survival in malignant and non-malignant cells has provided an unprecedented opportunity to target this pathway using BH3 mimetics for efficacious cancer treatment. BH3 mimetics provide great opportunities to treat hematological malignancies either as single agent or in combined regimen with low toxicity. In solid tumors survival dependencies are less evident maybe due to the higher level of heterogeneity or phenotypical plasticity. Combining chemotherapies with BH3 mimetics often provides a better tumor response in numerous preclinical studies but remains to be confirmed in clinical trials.

BH3 mimetics allowed to explore the full spectrum of MOMP biological effects. Related studies have shown that mitochondrial apoptosis and alternative cell death pathways are entwined. In retrospect, an alternative interpretation of the seminal observation made by Letai and colleagues in 2011 (that predisposition to MOMP prior treatment correlates with chemotherapeutic efficiency [31]) may be given. The effectiveness of chemotherapeutic treatments may indeed rather rely on MOMP induction than only on canonical apoptosis of tumor cells.

A growing body of evidence indicates that during treatments committed cells can actively contribute to the expansion and prolongation of therapy-induced effects through apoptotic spreading within the tumor as well as immune response activation and maintenance over time. Therapeutic induction of MOMP should therefore not be evaluated on the sole basis of its effect on each individual cancer cell apoptotic onset, but also on its ecosystemic effect. Arguably, caspase activation may provide unsuitable secondary effects by preventing inflammation and mask BH3 mimetic efficiency. This should be considered by designing therapeutic strategies that trigger death onset in the appropriate cell types but also rigorously influences the mode of cell death executed. To reach this objective, the role of the BCL-2 proteins and of MOMP in shaping progression and response to treatment in tumor ecosystems need further investigation to better unravel the intricate and possibly actionable communications involved, including during treatment where adaptative resistance may install.

## Figures and Tables

**Figure 1 biomolecules-10-01109-f001:**
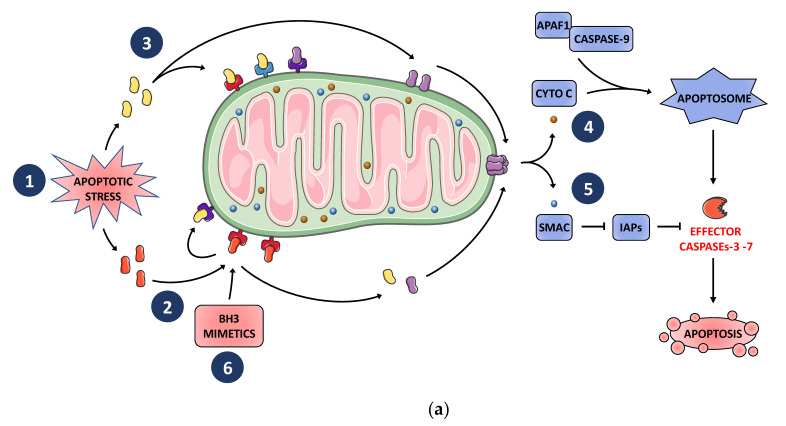
BCL-2 family proteins regulate MOMP through hierarchical interactions. (**a**) Upon apoptotic stress, **1.** BH3-only sensitizers NOXA and BAD bind to MCL-1 or BCL-2/BCL-XL, respectively, releasing BH3-only activators **2.** and BH3-only activators (BIM, PUMA or tBID) bind to anti-apoptotics (mainly BCL-2, BCL-xL, or MCL-1) or when in excess to apoptotic effectors BAX/BAK **3.** Activated BAX/BAK then form pores in MOM allowing cyto-c release in cytosol and subsequent apoptosome formation and caspase activation. **4.** The IAP inhibitor SMAC is also released during MOMP, indirectly regulating apoptotic caspase activity. **5.** BH3 mimetics compete with BH3-only proteins or BAX/BAK to bind anti-apoptotics. **6.** (**b**) BCL-2 family proteins preferentially bind to distinct members. For example, NOXA preferentially binds to MCL-1 or MCL-1 to BAK and BCL-2 to BAX.

**Figure 2 biomolecules-10-01109-f002:**
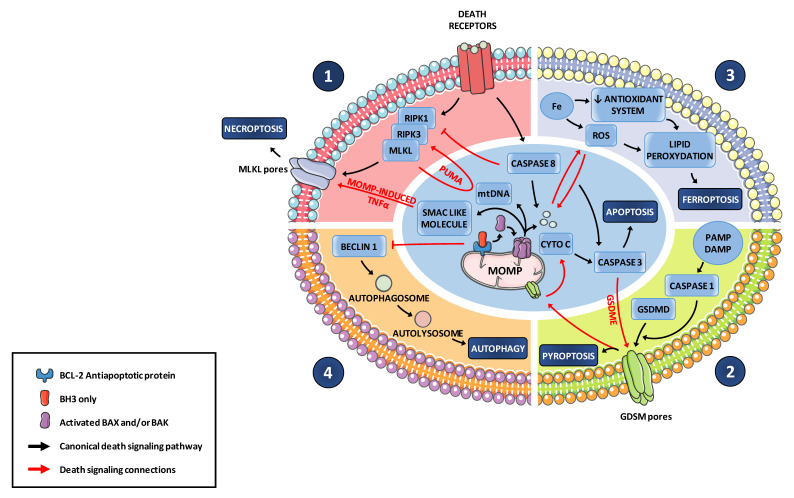
Death signaling crosstalk in tumor cells upon apoptotic caspase activation. **1.** Caspase-8 inhibits RIPK1-dependent necroptosis onset. In a caspase deficient context, MOMP-induced release of SMAC-like molecules promotes TNFα-dependent necroptosis. In turn, necroptosis-induced NF-κB-dependent transcription of PUMA promotes MOMP and mtDNA release. **2.** Pore-forming GSDME cleavage by activated caspase-3 connects apoptosis to pyroptosis. Reciprocally, pyroptosis leads to mitochondrial damages through GSDMD accumulation and mPTP-dependent release of ROS release and apoptosis activation. **3.** MOMP coincident release of ROS participates in ferroptosis whereas Fe-dependent ROS contributes to cyto-c release. **4.** Apoptosis interacts with autophagy signaling via Beclin 1-BCL-2 family anti-apoptotic proteins interactions.

**Figure 3 biomolecules-10-01109-f003:**
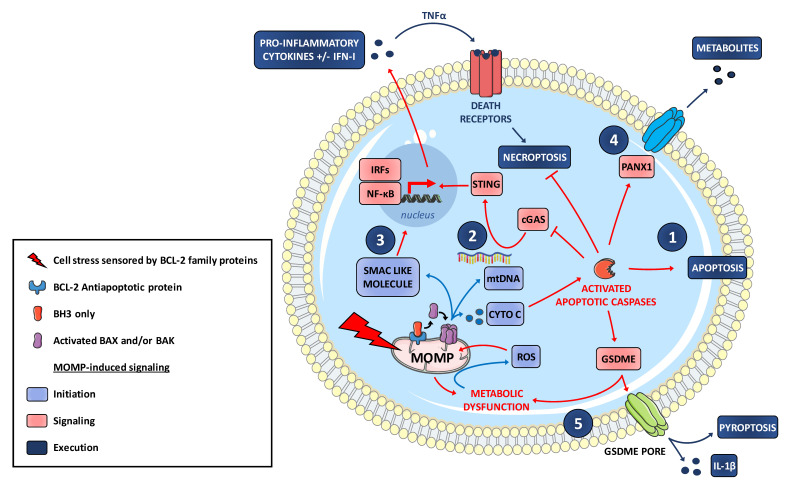
MOMP-induced signaling activation in tumor cells. MOMP-dependent release of factors to cytosol engages various signaling pathways. **1.** Cyto-c release leads to apoptotic caspase activation and subsequent apoptosis onset. **2.** MOMP-induced mtDNA release activates the cGAS-STING pathway resulting in a pro-inflammatory secretory phenotype when apoptotic caspases are inhibited **3.** Release of SMAC like molecules during MOMP promotes NF-κB activation. MOMP-induced TNFα can promote necroptosis in a context of caspase inhibition. **4.** Caspase-mediated opening of pannexin-1 channels at the plasma membrane facilitate the release of a select subset of metabolites. **5.** Pyroptosis-induced gasdermin D (GSDME) cleavage triggers mitochondrial metabolic dysfunctions and subsequent ROS accumulation that in turn promotes MOMP-dependent apoptosis.

**Figure 4 biomolecules-10-01109-f004:**
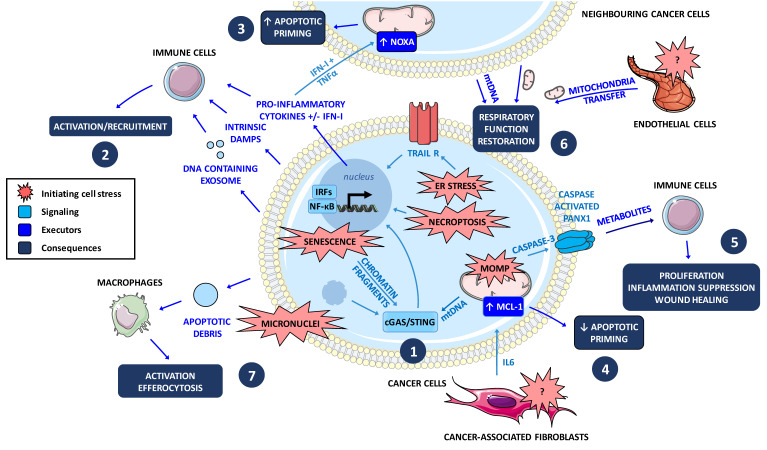
Intrinsic cancer cell response and intratumoral dialogue induced by chemotherapies. **1.** Senescence-dependent chromatin fragments, antimitotic-induced micronuclei or MOMP-released mtDNA contribute to proinflammatory secretory phenotypes in cancer cells treated by chemotherapies. Unprotected self DNA activate the cytosolic DNA sensor pathway cGAS-STING and downstream NF-kB and IFN-I signaling triggering chemo-induced secretome production. Chemo-induced necroptosis or ER stress (through intrinsic ligand-independent signaling) also promote a secretory phenotype. **2.** Pro-inflammatory cytokines, interferons as well as intrinsic (constitutive or inducible) Damage Associated Molecular Patterns (DAMPs) or DNA-containing exosomes can activate and recruit immune cells. **3.** Antimitotic-dependent secretome increases NOXA expression and apoptotic priming via TNFα/IFN-I in cancer cells in a paracrine manner. **4.** Pro-inflammatory cytokines (IL-6) from stromal cells (CAF) reduce apoptotic priming in cancer cells through IL-6-dependent MCL-1 upregulation. **5.** Active caspase-3 opens PANX1 channel releasing metabolites that potentially contribute to proliferation, inflammation suppression, and wound healing phenotype in neighboring healthy cells. **6.** Mitochondria or mtDNA transfer from endothelial cell to cancer cell or between two cancer cells can repopulate mitochondrial network in committed cells and restore respiratory functions. **7.** Apoptotic debris activate efferocytosis and pro-inflammatory cytokine secretion by macrophages.

**Table 1 biomolecules-10-01109-t001:** BH3 mimetics clinically used.

BH3 Mimetic *	Target	Condition
BH3 Mimetic Approved for Clinical Use		
Venetoclax	BCL-2	Chronic lymphocytic leukemia
BH3 Mimetics under Clinical Investigation		
Venetoclax	BCL-2	Hematologic malignancies Solid cancers
APG2575	BCL-2	Hematologic malignancies
S65487	BCL-2	Relapsed/refractory hematologic malignancies
BGB11417	BCL-2	Mature B-cell malignancies
AZD5991	MCL-1	Hematologic malignancies
S64315	MCL-1	Hematologic malignancies
AMG176	MCL-1	Relapsed/refractory hematologic malignancies
AMG397	MCL-1	Hematologic malignancies
ABBV467	MCL-1	Multiple myeloma
APG1252	BCL-xL, BCL-2	Lung cancers
AZD0466	BCL-xL, BCL-2	Hematologic malignancies Solid cancers
Navitoclax	BCL-xL, BCL-2, BCL-W	Hematologic malignancies Solid cancers
AT101	BCL-xL, BCL-2, MCL-1	Hematologic malignancies Solid cancers

* Many BH3 mimetics, selective or targeting several anti-apoptotic proteins, are currently under clinical investigation for the treatment of hematologic malignancies and solid cancers. In most cases, they are combined with chemotherapies. These data were extracted from https://clinicaltrials.gov website on April 2020. BCL-2 inhibitor venetoclax is the only BH3 mimetic approved by the FDA for treating chronic lymphocytic leukemia.

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
