# Peer review of "Targeting of BCL-2 Family Members during Anticancer Treatment: A Necessary Compromise between Individual Cell and Ecosystemic Responses?"

_biomolecules, 2020, doi:10.3390/biom10081109_

Round 1
Reviewer 1 Report
Summary: This verbous review article begins with general overviews of Bcl-2 family and BH3 mimetic-related apoptosis. It shifts towards MOMP and potential cell death cross-talk pathways, with highlights of inflammation and immune response to MOMP (unclear ties to Bcl-2 family specifically). Overall it reads very densely and feels like two reviews combined: one for Bcl-2 and another for MOMP/cell death pathways.
Comments:
- Typo in abstract “compouds”should be compounds
- Line 58 What is viral mimicking inflammation vs other inflammation – assume not general knowledge
- Line 65 those instead of these
- Line 67 refers to many studies yet there is only one reference – original texts instead of a review should be referenced
- Line 72, “It is understood as one common mechanism” clarify “It” as the mechanism – BLC2 family alteration in general? The previous sentence gave detailed examples
- Line 82 discriminating from not to
- Several English grammar issues throughout text
- Use of < > vs “ ” be consistent, most of the usage is unnecessary as activate/activator are common terms in the BCL2 field
- Define the players early: sensitizer vs activator; do not use quotations
- Section 2 is very wordy – would benefit from a table that outlines the anti- vs pro-apoptotic BLC2 proteins and their preferred affinities. This would greatly reduce and clarify the text for the readers.
- Reference for MOMP players Lines 123-131
- Check reference style for journal: Is et al. allowed for reference style for this journal? Some references include this after 10 authors listed
- Several long sentences that could be separated e.g. Lines 220-226
- Targeting the Bcl-Xl/Bcl-2 interface in combination with Noxa inducing compounds has been published by Alan Eastman’s group and should be referenced in section 3
- Lines 294-295 need rewording, confusing as written
- Line 296 – IAP should be in ( )
- Section 4 should be dedicated to clinical results as the subtitle indicates yet many examples and references included are pre-clinical studies (cell lines, animal models)
- Such as lines 292-298 (not quite BH3 mimetics but certainly not clinical)
- Line 353: you didn’t establish anything – this is a review. Your reference indicates the higher the Bcl-Xl expression, the more drug needed to inhibit this target; your conclusion does not make sense as written.
- “minority” MOMP does not make sense. Low levels of MOMP or low fraction of mitochondria undergoing MOMP
- Line 367 reference incomplete
- Statements utilizing Peco et al 2016 are muddy in section 5. “Evidenced” is incorrectly used.
- Failures of BH3 mimetic efficacy: Incomplete MOMP (discussed) vs unprimed cells (mentioned in a different section) vs non-reliance on BH3 targeted protein or reliance on several anti-apoptotic proteins at once – all must be hit in order to kill
- Line 454 missing a word – whether these can be triggered….to be elucidated
- Discussion of caspase activation as detrimental to anti-cancer treatment seems out of place in a review paper of apoptosis-inducing caspase-dependent drugs. There is potential for crosstalk among various cell death pathways, but which ones have evidence in the Bcl2 family realm? Just because pathways can have crosstalk, does not mean they do in the clinical setting.
- Inflammation and immune response are discussed in a Bax/Bak independent context and seem beyond the scope of this review. All of Section 6 seems out of scope.
- Line 561 typo “chekpoint” should be checkpoint
- Table 2 (too small to read text when printed, edges cut off): it is not inherently clear why we should care about caspase 7 in the context of Bcl2 family proteins/death. This seems tangental. What value does this table add to the reader’s knowledge?
- It appears that not all MOMP is the same? Which of the extensive signalizing pathways downstream of MOMP relate to Bcl-2/Xl/Mcl-1 inhibition?
- Failing to induce full caspase activation in all cells vs only a subset of cells undergoing full MOMP/caspase activation while the remainder of tumors are resistant with no MOMP/caspase activation at all. Clinically both are considered failures but can you truly state that one is more likely than the other in the clinical setting? As written failure for full caspase activation is THE failure of BH3 mimetics –which is not true. Lack of priming, reliance on multiple anti-apoptotic proteins, lack of sufficient drug to inhibit target, etc. all may contribute to incomplete clinical responses.
- Line 620 unclear why Figure 1 is referenced here – there are no “SAMPS” in your Figure 1
- Line 683 refers in to impact on neighboring cells, incorrect figure is referenced
- Line 692 “theses”?
- Line 704 remove the word “which”
- Line 727 missing (
- Overall the text is verbose and hard to read. Many important facts feel buried within the text. Readers would benefit from brevity and clarity.
Author Response
REVIEWER 1
We thank reviewer 1 for his/her careful reading
Comments:
1. Typo in abstract “compouds”should be compounds OK
2. Line 58 What is viral mimicking inflammation vs other inflammation – assume not general knowledge :
This is described ligne 555 with corresponding references
3. Line 65 those instead of these : OK
4. Line 67 refers to many studies yet there is only one reference – original texts instead of a review should be referenced : Beroukhim 2010 is not a review, 2 original reports have beeb added (Beverly 2009
(MYC-induced myeloid leukemogenesis is accelerated by all six members of the antiapoptotic BCLfamily and Zhou 2001 MCL1 transgenic mice exhibit a high incidence of B-cell lymphoma manifested as a spectrum of histologic subtypes
5. Line 72, “It is understood as one common mechanism” clarify “It” as the mechanism – BLC2 family alteration in general? The previous sentence gave detailed examples : This is understood as…corrected
6. Line 82 discriminating from not to : OK
7. Several English grammar issues throughout text : We have carefully read and corrected the text
8. Use of < > vs “ ” be consistent, most of the usage is unnecessary as activate/activator are common terms
in the BCL2 field .Define the players early: sensitizer vs activator; do not use quotations : « » deleted
9. Section 2 is very wordy – would benefit from a table that outlines the anti- vs pro-apoptotic BLC2 proteins and their preferred affinities. This would greatly reduce and clarify the text for the readers. We have now included the new figure 1 depicting BCL-2 homologues at MOM and their preferential affinities.
10. Reference for MOMP players Lines 123-131 : Juin 2013, added
11. Check reference style for journal: Is et al. allowed for reference style for this journal? Some references
include this after 10 authors listed : OK (Vancouver Style)
12. Several long sentences that could be separated e.g. Lines 220-226 : Style editing has been improved
13. Targeting the Bcl-Xl/Bcl-2 interface in combination with Noxa inducing compounds has been published by Alan Eastman’s group and should be referenced in section 3 : Albershardt 2011 added
14. Lines 294-295 need rewording, confusing as written : modified
15. Line 296 – IAP should be in ( ) : 0K
16. Section 4 should be dedicated to clinical results as the subtitle indicates yet many examples and references included are pre-clinical studies (cell lines, animal models) Such as lines 292-298 (not quite BH3 mimetics but certainly not clinical) : « Preclinical » added in title of the section
17. Line 353: you didn’t establish anything – this is a review. Your reference indicates the higher the Bcl- Xl expression, the more drug needed to inhibit this target; your conclusion does not make sense as written. We use « This might result from» instead of established
18. “minority” MOMP does not make sense. Low levels of MOMP or low fraction of mitochondria undergoing MOMP : minority MOMP is the original expression used by Ichim et al (2015). It has now been replaced by the proposed definition
19. Line 367 reference incomplete : OK, corrected
20. Statements utilizing Peco et al 2016 are muddy in section 5. “Evidenced” is incorrectly used. : we now use suggest instead of evidenced
21. Failures of BH3 mimetic efficacy: Incomplete MOMP (discussed) vs unprimed cells (mentioned in a different section) vs non-reliance on BH3 targeted protein or reliance on several anti-apoptotic proteins
at once – all must be hit in order to kill : We partially re-write this section to clarify each point of failures of BH3 mimetics
22. Line 454 missing a word – whether these can be triggered….to be elucidated : Now added
23. Discussion of caspase activation as detrimental to anti-cancer treatment seems out of place in a review paper of apoptosis-inducing caspase-dependent drugs. There is potential for crosstalk among various
cell death pathways, but which ones have evidence in the Bcl2 family realm? Just because pathways can have crosstalk, does not mean they do in the clinical setting. We feel that preclinical experimental evidence should be considered and as caspase inhibitors are currently in clinical trial, this has to be mentioned in an up to date review !
24. Inflammation and immune response are discussed in a Bax/Bak independent context and seem beyond the scope of this review. All of Section 6 seems out of scope. Since BCL2 family regulates MOMP and
MOMP is critical in mitochondrial DNA release (involving BAX macropores !) and downstream inflammatory signaling, we believe that in 2020 these new data/concepts should be included in a review about BCL2 family. This point is of particular importance for possible short or long term side efffects
when manipulating BCL2 family in humans.
25. Line 561 typo “chekpoint” should be checkpoint : OK corrected
26. Table 2 (too small to read text when printed, edges cut off): it is not inherently clear why we should care about caspase 7 in the context of Bcl2 family proteins/death. This seems tangental. What value does this table add to the reader’s knowledge? RPPA data include active caspase-7 but not active caspase-3 data. We thus infer in our computational analysis that active casp-7 results overlap active casp-3. We have now deleted Table 2 and adapted corresponding text.
27. It appears that not all MOMP is the same? Which of the extensive signalizing pathways downstream of MOMP relate to Bcl-2/Xl/Mcl-1 inhibition? In controling MOMP, BCL2 family contributes to downstream inflammatory signaling and this point (as mentioned above in point 24) shoul be considered to therapeutic use of BH3 mimetics in cancers.
28. Failing to induce full caspase activation in all cells vs only a subset of cells undergoing full MOMP/caspase activation while the remainder of tumors are resistant with no MOMP/caspase activation at all. Clinically both are considered failures but can you truly state that one is more likely
than the other in the clinical setting? As written failure for full caspase activation is THE failure of BH3 mimetics –which is not true. Lack of priming, reliance on multiple anti-apoptotic proteins, lack of sufficient drug to inhibit target, etc. all may contribute to incomplete clinical responses. We have
now moderated our purpose and noted: possibly failing to induce full caspase activation (See also point 21)
29. Line 620 unclear why Figure 1 is referenced here – there are no “SAMPS” in your Figure 1 : OK
30. Line 683 refers in to impact on neighboring cells, incorrect figure is referenced : OK
31. Line 692 “theses”?: OK corrected
32. Line 704 remove the word “which” : corrected
33. Line 727 missing ( : OK
34. Overall the text is verbose and hard to read. Many important facts feel buried within the text. Readers would benefit from brevity and clarity. We made the choice in this review to stimulate our
understanding of BCL2 family including all these new concepts that have recently emerged (MOMP and inflammation, BH3 mimetics : use and resistance…). We have now deleted some sentences and improved style editing
Reviewer 2 Report
The review focuses on a treatment of cancer with inhibitors of antiapoptotic proteins of the Bcl-2 family that act as BH3 mimetic (that is they target the BH3-binding groove on antiapoptotic Bcl-2 family members and mimic the binding of BH3 domain of proapoptotic proteins). The manuscript summarises the effects of these substances on cells and put it in the context of what is known on the regulation of cell death (apoptosis, but also proptosis, ferroptosis) in cancer cells.
I appreciate that authors had written the review on this topic. To my knowledge, the text is well written and follows the scientific logic.
The manuscript, though, appears not to be thoroughly edited. It almost looks like authors, by mistake, submitted the pre-final version of the manuscript. Some sentences are either grammatically incorrect or too complicated that they are difficult to understand. There are numerous mistakes in punctuation and text formatting. Some of the references are incomplete. There are typos in Tab. 1, Tab 2 does not display properly. Some abbreviations are used but not explained. e.g. AML (acute myeloid leukemia).
I recommend to accept the manuscript for publication in Biomolecules, but only after the thorough editing of the text.
Author Response
REVIEWER
2
The review focuses on a treatment of cancer with inhibitors of antiapoptotic proteins of the Bcl-2 family that act as BH3 mimetic (that is they target the BH3-binding groove on antiapoptotic Bcl-2 family members and mimic the binding of BH3 domain of proapoptotic proteins). The manuscript summarises the effects of these substances on cells and put it in the context of what is known on the regulation of cell death (apoptosis, but also proptosis, ferroptosis) in cancer cells.
I appreciate that authors had written the review on this topic. To my knowledge, the text is well written and follows the scientific logic.
The manuscript, though, appears not to be thoroughly edited. It almost looks like authors, by mistake, submitted the pre-final version of the manuscript. Some sentences are either grammatically incorrect or too complicated that they are difficult to understand. There are numerous mistakes in punctuation and text formatting. Some of the references are incomplete. There are typos in Tab. 1, Tab 2 does not display properly.
Some abbreviations are used but not explained. e.g. AML (acute myeloid leukemia).
I recommend to accept the manuscript for publication in Biomolecules, but only after the thorough editing of the text.
We thank the reviewer 2 for his (her) postive comments. We carefully have reedited the manucript (including AML), as mentioned above.
Reviewer 3 Report
The review by Barillé et al provides an extensive overview of Bcl-2 family mode(s) of action, and their implication from seemingly every possible angle in cell death processes. The manuscript takes into account arguments coming both from landmark papers from some 15 years ago, as well as very recent ones. Basic science as well as clinical data are assembled, which produces a remarkable piece of work.
All the essential concepts regarding Bcl-2 family members, how they operate and how they can or could be targeted in cancer (at the cellular level or at the organism level) are included. The review also features very “trendy” considerations regarding the communication between cancer cells and the stroma and how the dialog between all the cell types emerges as a therapeutic target. I can only congratulate the authors for the quality of the work they assembled.
On the overall structure of the manuscript, I would only remind the authors that reviews are very often how newcomers enter a field; not only confirmed scientists, but also students. With this in mind, I would like to encourage the authors to avoid extremely long sentences which might blur the message and deter students from an otherwise excellent review. For the sake of clarity, may be the authors could include more sub-divisions within the chapters? This remark goes essentially for the beginning of the manuscript which does not read as easily as the rest of it.
The figures are very well designed and clear. I would recommend on Figure 1 a blow up the scheme and to keep the box at the same size. Table 2 should be provided as “landscape” instead of “portrait”.
Please find below some typo errors that have been noted, and some suggestions of modifications.
Line 23: “BH3 mimetics administration (…) could instead allow death signal production by “non completely dead” cell populations”.
I understand that cells can execute a death program in an asynchrounous manner, and that this lack of synchronicity might increase BH3 mimetics performance. May be rephrase and use “non terminally committed dying cells”.
Line 17: a growing body of evidence indicates
Line 75: distant metastasis: pleonasm?
Line 82: discriminating FROM
Line 91: cytochrome c (indicate that it will thereon be abbreviated to cyto-c)
Lines 94-95: we suggest to replace the double negation by a simpler formulation for the sake of clarity.
Line 105: sensitizer (remove S) proteins.
Line 106: “anti-apoptotic load” does not read right. Do the authors merely mean that binding of sensitizer proteins relieves/releases BAX/BAK from anti-apoptotic inhibition?
Lines 113-118: the discussion about how selective the interactions are between Bcl-2 family members due to their differential affinities is much appreciated, and how this translate into remodeling the network of interactions when one partner is absent in cancer cells is a very valid point. But the fact that all the Bcl-2 proteins are not expressed in all the tissues is certainly another parameter to modulate the selectivity of the interactions in normal cells. Could the authors include a note about how this may (or may not?) butt in the equation?
Line 118: Even though
Line 129: several hundreds of
Line 161: aN individual lack of priming: this “individual” concept is unclear. Could the authors explain better?
Line 175: paved the way FOR the development…
Line 178: to the development of molecules that, BY inhibiting prosurvival…
Line 180: into the BH3 binding pocket OF the prosurvival
Line 202: the verb is missing “Remarkable response in patients with CLL even those whoSE have failed standard chemoimmunotherapy”.
Line 246-247: why not writing the names of the Bcl-xL inhibitors identified by Lessene?
Line 253 and 256: MCL-1
Line 294: and therefore RATHER SPARES platelet-expressed Bcl-xL?
Line 318: affecting only THOSE more primed for apoptosis, like IN THE hemapopoietic system…?
Line 332: between different distinct cell types.
Line 360: mitophagy was going on.
Line 262: in cancer cells
Line 367: no year associated to the citation.
Line 370: shuttling between mitochondria AND cytosol
Line 380: dependence on MCL-1 in treated cancer cellS
Line 392: pro-apoptotic counterparts THROUGH their BH3-binding site but also THROUGH other interfaces ?
Line 393: BCL-2 or
Line 442: an inflammatory
Line 454: and HOW all these can be
Lines 549-551: “Exploration of public databases comprising matched proteomic and RNA expression data indicate that this is not always observed.” Is there a reference associated, or is this a personal computation of the authors?
Author Response
REVIEWER 3
The review by Barillé et al provides an extensive overview of Bcl-2 family mode(s) of action, and their implication from seemingly every possible angle in cell death processes. The manuscript takes into account arguments coming both from landmark papers from some 15 years ago, as well as very recent ones. Basic science as well as clinical data are assembled, which produces a remarkable piece of work.
All the essential concepts regarding Bcl-2 family members, how they operate and how they can or could be targeted in cancer (at the cellular level or at the organism level) are included. The review also features very
“trendy” considerations regarding the communication between cancer cells and the stroma and how the dialog between all the cell types emerges as a therapeutic target. I can only congratulate the authors for the quality of
the work they assembled.
We thank the reviewer 3 for his/her positive comments
On the overall structure of the manuscript, I would only remind the authors that reviews are very often how newcomers enter a field; not only confirmed scientists, but also students. With this in mind, I would like to encourage the authors to avoid extremely long sentences which might blur the message and deter students from an otherwise excellent review. For the sake of clarity, may be the authors could include more sub-divisions within the chapters? This remark goes essentially for the beginning of the manuscript which does not read as easily as the rest of it.
The figures are very well designed and clear. I would recommend on Figure 1 a blow up the scheme and to keep the box at the same size. Table 2 should be provided as “landscape” instead of “portrait”. Table 2 has been
deleted
Please find below some typo errors that have been noted, and some suggestions of modifications.
Line 23: “BH3 mimetics administration (…) could instead allow death signal production by “non completely
dead” cell populations”. OK
I understand that cells can execute a death program in an asynchrounous manner, and that this lack of
synchronicity might increase BH3 mimetics performance. May be rephrase and use “non terminally committed
dying cells”. OK
Line 17: a growing body of evidence indicates : OK
Line 75: distant metastasis: pleonasm? OK
Line 82: discriminating FROM : OK
Line 91: cytochrome c (indicate that it will thereon be abbreviated to cyto-c) : OK
Lines 94-95: we suggest to replace the double negation by a simpler formulation for the sake of clarity. : OK
Line 105: sensitizer (remove S) proteins. : OK
Line 106: “anti-apoptotic load” does not read right. Do the authors merely mean that binding of sensitizer
proteins relieves/releases BAX/BAK from anti-apoptotic inhibition? We meant « apoptotic load » : now
corrected
Lines 113-118: the discussion about how selective the interactions are between Bcl-2 family members due to their differential affinities is much appreciated, and how this translate into remodeling the network of
interactions when one partner is absent in cancer cells is a very valid point. But the fact that all the Bcl-2 proteins are not expressed in all the tissues is certainly another parameter to modulate the selectivity of the interactions in normal cells. Could the authors include a note about how this may (or may not?) butt in the equation?
This point is important and we have now added on line 168: Importantly expression of BCL-2 proteins varies in normal tissues where apoptotic priming is developmentally regulated (Sarosiek et al., 2017)
Line 118: Even though : corrected
Line 129: several hundreds of : corrected
Line 161: aN individual lack of priming: this “individual” concept is unclear. Could the authors explain better?
We have deleted « individual » and added in tumors for better understanding
Line 175: paved the way FOR the development… : OK
Line 178: to the development of molecules that, BY inhibiting prosurvival…OK
Line 180: into the BH3 binding pocket OF the prosurvival : OK
Line 202: the verb is missing “Remarkable response in patients with CLL even those whoSE have failed
standard chemoimmunotherapy”. : OK
Line 246-247: why not writing the names of the Bcl-xL inhibitors identified by Lessene? WEHI-539 added
Line 253 and 256: MCL-1 : OK
Line 294: and therefore RATHER SPARES platelet-expressed Bcl-xL? : OK
Line 318: affecting only THOSE more primed for apoptosis, like IN THE hemapopoietic system…? : OK
Line 332: between different distinct cell types. « different « removed
Line 360: mitophagy was going on. : OK
Line 262: in cancer cells : OK
Line 367: no year associated to the citation. : OK
Line 370: shuttling between mitochondria AND cytosol : OK
Line 380: dependence on MCL-1 in treated cancer cellS : OK
Line 392: pro-apoptotic counterparts THROUGH their BH3-binding site but also THROUGH other interfaces ?
OK
Line 393: BCL-2 or OK
Line 442: an inflammatory : OK
Line 454: and HOW all these can be : OK
Lines 549-551: “Exploration of public databases comprising matched proteomic and RNA expression data
indicate that this is not always observed.” Is there a reference associated, or is this a personal computation of the
authors? Yes (« personal computation » added in the text)
Round 2
Reviewer 1 Report
The revisions have improved the readability and clarity of the manuscript.
All figures could be enlarged with larger text to make them easier to see and comprehend the extensive signaling networks reviewed. Figure 1: please do not use yellow lines in part b (hard to see - a darker color would be beneficial), also PUMA is lacking association to BAK/BAX binding.
Line 587 Unclear why referencing Figure 2 - it does not contain inflammation or SAMPS
Line 653 Unclear why referencing Figure 3
Several typos can be found:
Line 67 missing )
Line 286 "do resist to", simply "resist" would be more clear
Line 324 either incorrect "." after reference or lack of capitalization of next sentence
Line 446 defines type I-IFN but the abbreviation is used prior (lines 417, 433), also this term sometimes has the hyphen and sometimes does not within the text, and sometimes is type I-IFN and is sometimes just I-IFN, keep it consistent throughout the text and figures.
Line 471 "in in"
Figure 3 legend line 490 "facilitatethe" needs a space
Line 549 "forster"??
Line 560 comma not needed
Line 610 "transfert"??
Line 678 "therfore"
Author Response
The revisions have improved the readability and clarity of the manuscript.
We thank again this reviewer for his thorough analysis of the manuscript
All figures could be enlarged with larger text to make them easier to see and comprehend the extensive signaling networks reviewed. Figure 1: please do not use yellow lines in part b (hard to see - a darker color would be beneficial), also PUMA is lacking association to BAK/BAX binding.
Typos were changed from 10 to 12 in size and colors were changed as requested.
Line 587 Unclear why referencing Figure 2 - it does not contain inflammation or SAMPS
The reference to Figure 2 was deleted.
Line 653 Unclear why referencing Figure 3
This line now refers, more adequately, to Table 1
Several typos can be found:
All the following typos were corrected.
Line 67 missing )
done
Line 286 "do resist to", simply "resist" would be more clear
done
Line 324 either incorrect "." after reference or lack of capitalization of next sentence
"." was incorrect
Line 446 defines type I-IFN but the abbreviation is used prior (lines 417, 433), also this term sometimes has the hyphen and sometimes does not within the text, and sometimes is type I-IFN and is sometimes just I-IFN, keep it consistent throughout the text and figures.
done
Line 471 "in in"
done
Figure 3 legend line 490 "facilitatethe" needs a space
done
Line 549 "forster"??
done (foster)
Line 560 comma not needed
done
Line 610 "transfert"??
done (transfer)
Line 678 "therfore"
done (transfer)